# Instructional Coaching for Implementing Visible Learning: A Model for Translating Research into Practice

**Jim Knight**

Center for Research on Learning, The University of Kansas, Lawrence, KS 66045, USA;
jim@instructionalcoaching.com; Tel.: +1-805-336-5789

**Abstract:** Visible Learning has been one of the most influential research initiatives conducted in education in the past few decades, and at the same time, instructional coaching is becoming one of the most popular forms of professional development. This paper considers how the implementation of Visible Learning may be supported through instructional coaches by: (a) offering a brief summary of the central tenants of Visible Learning; (b) summarizing the foundational research on instructional coaching conducted at the Kansas Coaching Project at The University of Kansas Center for Research on Learning; (c) describing what those findings reveal about effective instructional coaching practices; and (d) pointing out how the research findings suggest that instructional coaching should be used to support the implementation of Visible Learning or any other educational innovations.

**Keywords:** instructional coaches coaching; visible Learning; translating research into practice; educational research

## 1. Introduction

One of the most formidable challenges facing educational leaders today is the challenge of translating research into practice. In the United States, billions of dollars are spent annually on professional development activities designed to provide the tools necessary for teachers to make evidence-based practices a central part of their teaching (determining the exact cost of professional development is difficult, if not impossible, but in 2015, one study estimated that the 50 largest districts in America spend eight billion dollars on professional development annually). Unfortunately, professional development often has little impact on what actually happens in classrooms, despite the money and effort expended [1].

This paper explores how this challenge may be addressed through the use of instructional coaching. Specifically, the paper will suggest how one widely implemented educational construct, Visible Learning [2–4], may be implemented through the use of instructional coaches.

Instructional coaches are increasingly serving as facilitators of professional learning, and given the widespread use of Visible Learning, exploring how implementation of this construct may be supported through instructional coaches seems relevant and important. This paper addresses this topic by: (a) offering a brief summary of the central tenants of Visible Learning; (b) summarizing the foundational research on instructional coaching conducted at the Kansas Coaching Project at The University of Kansas Center for Research on Learning; (c) discussing how the coaching process derived from those findings may be utilized to support the implementation of Visible Learning and other innovations; and (d) concluding with suggestions for future research on coaching and Visible Learning.

## 2. Visible Learning

Visible Learning is one of the most influential research initiatives conducted in the field of education in the past few decades. Around the world, educators are reading and learning about the highest-leverage factors for student achievement, as identified in Hattie's evolving lists of best practices [2–4].

This original work on Visible Learning was conducted over a 15- to 20-year period, and it "involved analyzing more than 800 meta-analyses composing around 80,000 studies in which an estimated . . . 250 million learners took part" [4]. As of 2018, approximately 600 additional studies had been conducted to further refine identification of the factors most likely to have a positive impact on student achievement.

To conduct their research, Hattie and his colleagues [2–4] identified approximately 150 factors suggested in the literature as having an impact on student achievement. Then, to determine which factors had the greatest impact, the researchers compared the effect sizes of all factors. Effect size is usually calculated by comparing the averages of two conditions; for example, classes where cooperative learning is not used by teachers and students vs. classes where cooperative learning is used by teachers and students. "The beauty of effect sizes," according to Hattie and Zierer [4], "is that, once computed, they can be reasonably compared across many interventions" (p. ix). The greater the effect size, generally speaking, the greater the positive impact on student achievement.

Hattie's findings have laid the foundation for many additional publications on related topics, including feedback [5] assessment [6] literacy [7] mathematics [7] and science [8]. This paper focuses on Visible Learning as it is described in Hattie and Zierer's most recent [2] comprehensive discussion, 10 Mindframes for Visible Learning: Teaching for Success. The 10 mindframes are as follows:

**1. Visible Learning is about teachers making their impact visible.** To make learning visible, teachers should use formative evaluation ($d = 0.90$) and consider employing Response to Intervention ($d = 1.07$). Hattie and Zierer [4] suggest that teachers make learning visible through a process that may be remembered by the somewhat gloomy acronym, DIE (the authors put this in more positive terms, saying that the acronym may be remembered by acknowledging that teachers are "to die for").

**Diagnosis**—understanding what each student brings to the lesson, his or her motivations, and willingness to engage.

**Intervention**—having multiple interventions, such that if one does not work with the student, the teacher changes to another. It also involves knowing high-probability interventions, recognizing when to switch, and not creating blame language about why the student is not learning.

**Evaluation**—knowing the skills, having multiple methods, and collaboratively debating the magnitude of impact from the interventions (p. 8).

**2. I see assessment as informing my impact and next steps.** After teachers gather data, they need to adapt their teaching so that more students learn. Simply put, teachers should "use assignments to clarify the following questions: Which of my goals did I achieve in the lesson? What material did I successfully get across to the learners? Which methods turned out to be useful for fostering learning? Which media were useful for fostering learning?" [4].

**3. I collaborate with my peers and my students about my conceptions of progress and my impact.** Professional learning is enhanced when teachers cooperate, exchange ideas, and share responsibilities. Hattie and Zierer [4] cite the results of a study by Eells [9] as evidence that collective efficacy has a high impact on student achievement ($d = 1.22$). They also claim that micro-teaching with video ($d = 0.88$) and professional development ($d = 0.51$) contribute to better professional learning, and consequently higher student achievement.

**4. I am a change agent and believe all students can improve.** According to Hattie and Zierer [4], "if the student is not learning, it is because we have not yet found the strategy to make learning happen. Successful learning . . . is the responsibility of all around the learner . . . [and] this necessarily involves seeing oneself as a change agent" (p. 40). To that end, the authors suggest that teachers employ different classroom management strategies ($d = 0.52$), "use preventive strategies," and "try to get a critical mass of learners to believe in . . . [their] . . . visions" (p. 57).

**5. I strive for challenge and not merely "doing your best."** Echoing Csíkszentmihályi's findings about engagement, Hattie and Zierer [4] contend that the best learning is just a little more challenging that the learner's current ability level. Work that is too hard is frustrating and may cause anxiety; work that is too easy is boring. Indeed, being boring has one of the most negative effect sizes of any factor studied by Hattie and his colleagues ($d = -0.49$). Factors related to this mindframe include teacher clarity ($d = 0.75$), goals ($d = 0.50$), and students skipping grades ($d = 0.68$). Students who have goals and who get timely feedback on their progress toward those goals are more likely to move towards what Csíkszentmihályi calls "a state of flow" and, therefore, more likely to achieve.

**6. I give and help students understand feedback, and I interpret and act on feedback given to me.** Feedback is one of the most powerful influences on student learning ($d = 0.75$). One way teachers can get and give feedback is through classroom discussion made possible by effective questioning ($d = 0.48$). Hattie and Zierer [4] list three major feedback questions: "Where am I going? How am I going? and Where to next?" (p. 80). The most powerful form of feedback," the authors explain, "is from students to teachers about their impact on the students" (p. 75).

**7. I engage as much in dialogue as monologue.** To foster better learning, teachers need to "get the balance right between their talking and explaining and listening and privileging student discussion" [4]. The authors report several studies that reveal that teaching is too often monological. For example, after reviewing 1500h of recordings, Clinton, Cairns, Mclaren, and Simpson (2014) found that teachers talk 89% of the time. To make classrooms more dialogical ($d = 0.82$), teachers should consider structuring lessons to include such learning structures as cooperative learning ($d = 0.59$), jigsaw ($d = 1.09$), and small-group learning ($d = 0.49$). Cooperative learning, according to Hattie and Zierer [4], is more effective when it is supported by direct instruction ($d = 0.59$).

**8. I explicitly inform students from the outset what successful impact looks like.** To succeed, students need to know what success will look like. For that reason, "The teacher needs to know, and the students informed, what success criteria of performance are to be expected and when and what students will be held accountable for from the lesson/activity" [4]. Teachers should formulate success criteria, clearly communicate the criteria, ensure students understand the criteria by asking them for feedback, use case studies to illustrate the criteria, and revisit the criteria to see what changes need to be made.

**9. I build relationships and trust so that learning can occur in a place where it is safe to make mistakes and learn from others.** Teacher-student relationships are an important part of effective teaching ($d = 0.72$). To increase the likelihood that students will succeed, teachers should reflect on their expectations ($d = 0.43$), avoid negative expectations, and reinforce students when they see them trying. Teachers should also pay attention to their language, use humor and cheerfulness, and strive to always be honest, fair, and credible.

**10. I focus on learning and the language of learning.** An essential part of this mindframe is recognizing how important it is to determine each student's prior knowledge, or learning level, and then to use that as the point of departure for new learning in the classroom. As part of that effort, teachers should try to assess students' self-efficacy, motivation, way of working, and conscientiousness. There are many ways to assess students' initial learning level that have high effect sizes. Teachers should also be aware of cognitive overload and ensure that assigned work is neither too challenging nor too easy.

Visible Learning sets forth a coherent and powerful model for improving student achievement. The challenge, as laid out in the introduction to this paper, is to find a methodology for translating the Visible Learning research and ideas into practice. The contention of this author is that instructional coaching offers a fitting methodology.

### 3. Instructional Coaching Research (All of the Articles Referenced Here Related to Research on Instructional Coaching May Be Downloaded at instructionalcoaching.com/research)

The instructional coaching model described below is the result of more than 20 years of systematic study. Research began in the years 1997–2003 with researchers interviewing more than 300 teachers

about their experiences with professional development. The findings were not positive. Interviews revealed that teachers: (a) had low expectations for professional development; (b) did not find that professional development met their needs; (c) complained that professional developers often failed to recognize the expertise that teachers already had; and (d) rarely implemented what they heard about in workshops [10].

Another study [11] explored professional developers' way of being, building on the finding that many teachers found the approach of professional developers to be a barrier to implementation. Specifically, the study compared a partnership approach to professional development (grounded in the principles of equality, choice, voice, reflection, dialogue, praxis, and reciprocity) with a traditional approach (based on direct instruction emphasizing fidelity of implementation. The principles were based on a synthesis of theories from education, business, psychology, sociology, cultural anthropology, and philosophy of science, in particular, the works of Bernstein, Block, Bohm, Eisler, Freire, and Senge [12–17].

The study focused on the engagement, happiness, learning, and expectation for implementation of participants in two workshops, one based on partnership principles and the other on fidelity of implementation. Participants in the former group were significantly more engaged and happier, produced higher scores on tests of what they learned, and reported they were 4.5 times (58–14) more likely to implement the reading strategy they had learned during the partnership approach than the one they had learned during the traditional approach [11].

Subsequently, funding from the U.S. Department of Education GEAR UP program supported several years of study of onsite professional development. The onsite professional developers were first described as learning consultants [18] then as instructional collaborators, and finally as instructional coaches [19]. During this time, the results of a number of informal studies allowed for development and refinement of what became an instructional coaching model (no model for anything similar to instructional coaching existed at that time). Although the preliminary studies lacked rigor [20], researchers were convinced that instructional coaching was a promising model for supporting implementation of evidence-based teaching strategies.

In 2007, researchers completed a more rigorous study of instructional coaching as it was being implemented by coaches studied by The University of Kansas Center for Research on Learning (KU-CRL) at the time. For this study, 51 teachers attended an after-school workshop on an inclusive teaching strategy, The Unit Organizer [21]. After the workshop, teachers were randomly assigned to two groups, one group that received coaching and another that did not receive coaching. As Figures 1–4 indicate, according to blind observations, the teachers who received coaching (a) were more likely to implement (87% to 33%); (b) taught with closer fidelity to the original model based on observations (7, 7, 6, 5, compared with 3, 3, 2, 1); (c) were more likely to continue to use the teaching routine (68% to 18%); and (d) reported higher expectations of using the teaching routine in the future (96% to 35%).

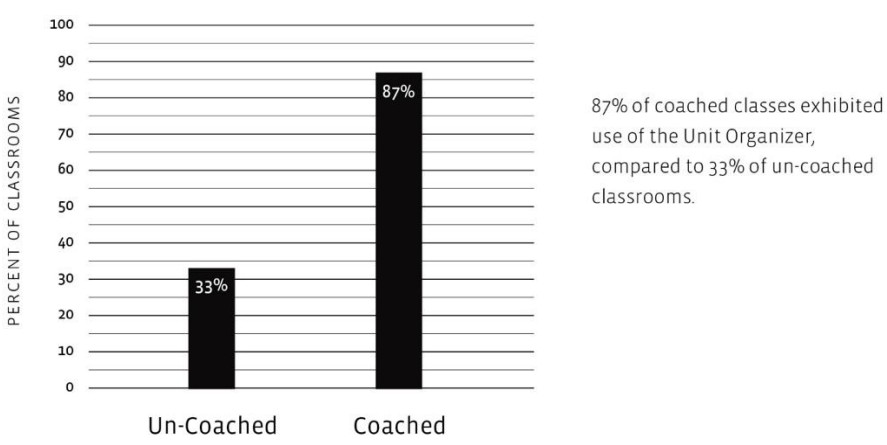

**Figure 1.** Implementation of the unit organizer [22].

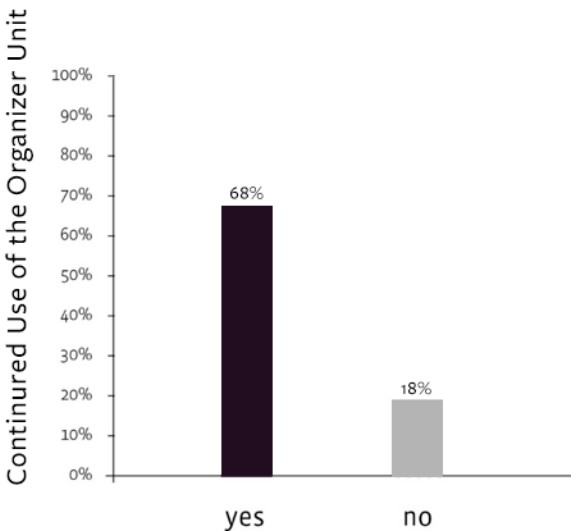

**Figure 2.** Continued use of the unit organizer [22].

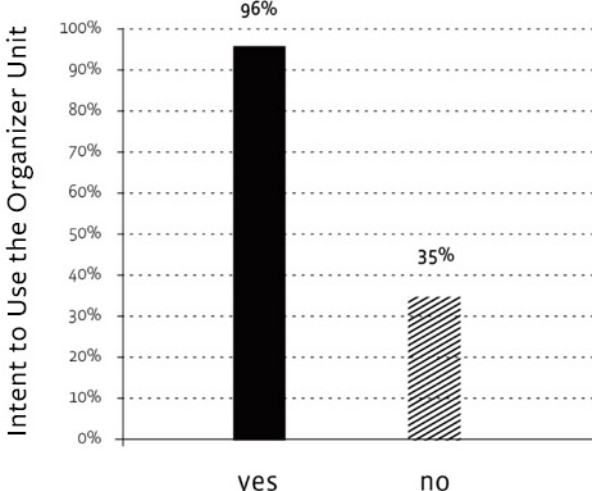

**Figure 3.** Intent to use the unit organizer [22].

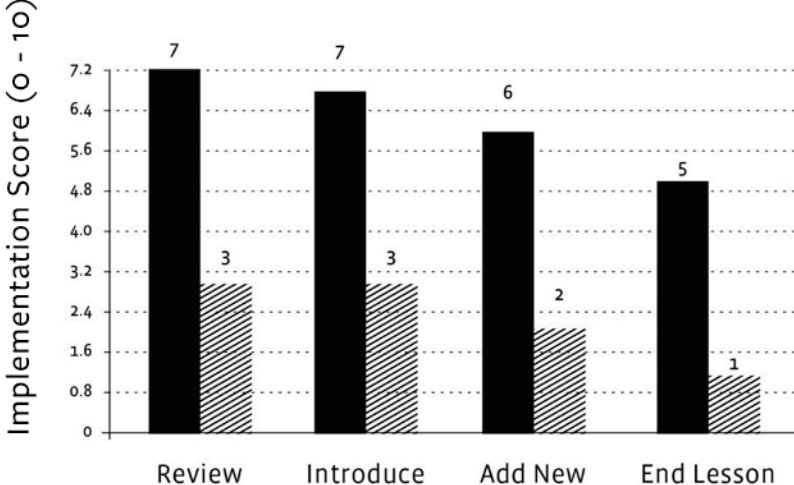

**Figure 4.** Fidelity of implementation [22].

Researchers at the Kansas Coaching Project utilized design research [23] to facilitate improvement and refinement of the instructional coaching model. Over time, the design methodology was modified

to create lean-design research (LDR), which combined design research with the research methods described by Ries in The Lean Startup: How Today's Entrepreneurs Use Continuous Innovation to Create Radically Successful Businesses [24]. See Figure 5 for an illustration of LDR.

Over five years, the instructional coaching process was significantly improved. Refinements included coaches and teachers using video to get a clear picture of reality, a revised model for goal setting (PEERS goals, described below), use of more effective questioning skills, implementation of a variety of ways to demonstrate teaching strategies, and a streamlined coaching process.

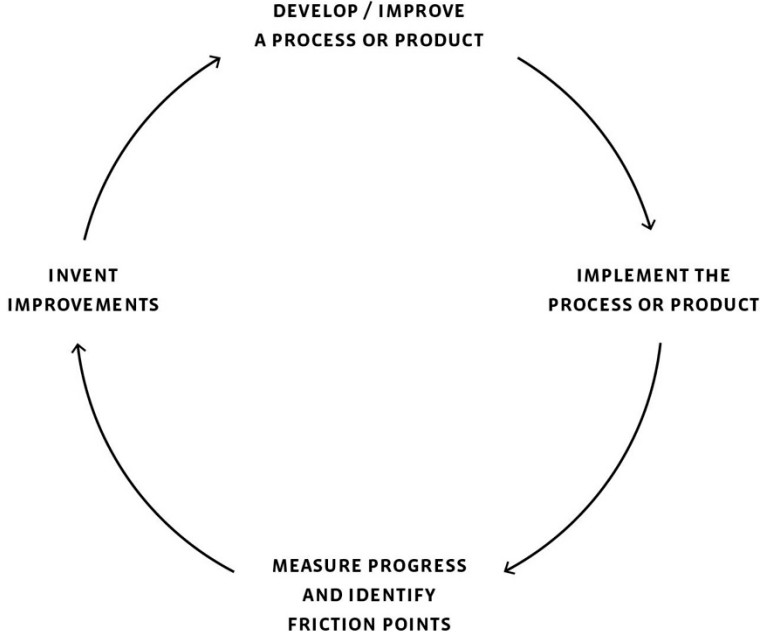

**Figure 5.** Lean-design research [25].

At the same time, researchers also conducted a multiple-baseline study of coaching to measure its impact on teaching and student engagement. Again, observation data showed that teachers used significantly more targeted teaching practices after they were coached than before. Additionally, measures of student engagement showed significant gains in time-on-task behavior after coaching ($d = 1.03$).

Other studies have been conducted over the past 20 years, including a review of the research literature [26] and a study of the characteristics of effective coaches [27]. Together, the studies described here and in various books related to instructional coaching [20,25,28–31] have led to the articulation of a simple model for instructional coaching, as described below.

## 4. Instructional Coaching Model

The research on instructional coaching conducted at the University of Kansas Center for Research on Learning has resulted in the development of a deceptively simple instruction coaching cycle. The cycle involves three elements: Identify, Learn, and Improve (see Figure 6).

During the Identify stage, coaches partner with teachers to identify a clear picture of reality, a PEERS goal, and a strategy the teacher will implement to hit the goal. During the Learn stage the coach helps prepare the teacher to hit the goal by clearly describing the strategy to be implemented, often with the help of a checklist, and then provides a model of the strategy in one or more ways. Finally, during the Improve stage, the coach supports the teacher as he or she makes adaptations until the goal is met.

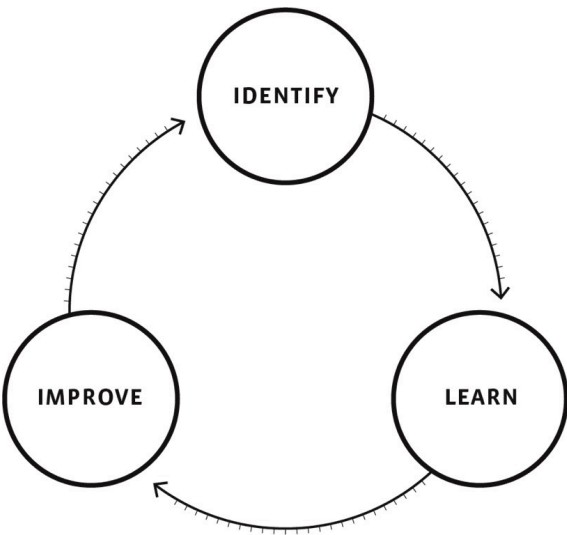

**Figure 6.** The impact cycle [25].

The definition of instructional coaching underlying the model is as follows: "Instructional coaches (a) partner with teachers to (b) analyze current reality, (c) set goals, (d) identify and explain teaching strategies to hit the goals, and (e) provide support until the goals are met" [25]. A description of each of these elements follows.

A fundamental tenet of instructional coaching is that, although described sequentially in this paper, it is not a simplistic one-size-fits-all formula for improvement. Rather, instructional coaches respond to the context in which coaching occurs, shaping what they do based on students' needs, teachers' insights, and other important factors. As such, the approach has been described as informed-adaptive [28].

Thus, informed coaches know a lot about the situation where coaching occurs. They have a deep understanding of individual teachers' strengths and concerns, and students' unique characteristics. They are emotionally intelligent, which means they are skilled at fostering trust and building relationships that are likely to lead to learning. Finally, informed coaches have a deep knowledge of instructional practices that enables them to offer more options to teachers who partner with them to meet students' needs.

Adaptive coaches respond to the unique contexts in which coaching occurs. Although instructional coaching, as described here, occurs within a framework, that framework is not a formula to be strictly followed, but rather a container for coaching conversations. Thus, each coaching conversation is individualized to a given context. That is, the questions coaches ask, the goals that are set, the teaching practices chosen by teachers, the way teachers learn and practice new strategies, and the modifications that are made are all unique to the partnership between teacher and coach. So, while instructional coaching involves a structure, in action it is individualized process, uniquely co-constructed by each coach and teacher.

For a construct such as Visible Learning to be implemented in a deep and effective way, a professional development model such as instructional coaching is essential. The rest of this paper explores what instructional coaching for Visible Learning would look like.

The following are components of the instructional coaching process that have emerged from the findings of the studies summarized above.

**Partnership approach**. A fundamental tenet of the instructional coaching model is that instructional coaches partner with teachers [11]. That is, they see themselves as equals with teachers, recognizing that every teacher brings expertise to a coaching conversation. As a result, instructional coaches do not tell teachers what to do, but at the same time, they do not withhold any expertise they have that might help a teacher meet a goal. Instructional coaches share ideas dialogically, balancing advocacy (explaining practices) with inquiry (asking questions and listening) [14] in a manner that

acknowledges that each teacher will likely need to modify practices to meet the unique needs of his or her students and to take advantage of his or her unique strengths as teachers. When they explain practices, instructional coaches are precise but provisional, always recognizing that teachers should and will have the last word on how practices are implemented.

**Application to Visible Learning**. Coaches who position teachers as partners need to employ a sophisticated approach to communication about Visible Learning. On the one hand, they must have a deep understanding of Visible Learning and be able to describe each of the elements of the framework clearly, otherwise, teachers will struggle to learn. At the same time, when taking the partnership approach, coaches must structure coaching so that teachers use their own knowledge and experience to decide how Visible Learning will be implemented. Coaches taking the partnership approach provide support when needed (perhaps suggesting ways in which goals may be measured or suggesting teaching strategies teachers might implement to reach goals) while also ensuring that teachers make the final decisions about what they will do in their classrooms. When coaches take the partnership approach, they ensure that teachers use their professional discretion to determine the best practices for their classroom.

### 4.1. Identify

**Identify a clear picture of current reality.** Once teachers are enrolled in coaching, the coaching process begins with getting a clear picture of reality. Because of perceptual errors, most people do not have a clear picture of what it looks like when they do what they do [30,32]. Therefore, before setting goals, teachers need to get a clear picture of what learning and teaching look like in their classrooms. If teachers set goals without a clear picture of reality, they may lack enthusiasm for the goal or choose a goal that does not address a top need in their classroom. Instructional coaches help teachers get a clear picture of reality by (a) video recording a lesson and sharing it with teachers; (b) interviewing students and sharing the interview data; (c) reviewing student work with the teachers; or (d) gathering objective data in the classroom and then sharing that data with the collaborating teacher [25].

**Application to Visible Learning.** Getting a clear picture of reality can occur in many ways and involve many aspects of Visible Learning; a few examples will be described here. To get a clear picture of reality, coaches can video-record lessons so that teachers can view the video to consider the level of engagement, the amount of monologue vs. dialogue, the effectiveness of questions asked, the clarity of instruction, or the amount of wasted, non-instructional time in their classroom. Coaches might also interview students to assess their learning level, motivation (by asking students to discuss their goals, strategies, and agency), or interest in learning.

Another important way in which coaches can help teachers get a clear picture of reality is by helping them use different tools for analyzing student work. This is most powerful when teachers have already developed learning outcomes, success criteria, and formative assessments of student learning, perhaps in an intensive learning team [28]. Coaches can also observe lessons, gather data, and share the data with teachers. Again, coaches could look for certain factors that are identified within Visible Learning as important contributors to student achievement. Essentially, coaches could gather many of the factors identified above in the section on video, including students' level of engagement, the amount of monologue and dialogue, the effectiveness of questions teachers ask, the clarity of instruction, and the amount of wasted, non-instructional time.

**Identify goals.** A second finding from our research over the past 20 years is that coaching is most effective when it is a goal-directed activity. As John Campbell, founder of Growth Coaching International (www.growthcoaching.org), has said, "if there's no goal, it is just a nice conversation" (personal communication, 2017). Teachers who implement new practices most effectively do so when they have clearly articulated goals. Utilizing lean-design research, we interviewed experts, reviewed the literature, and learned from coaches who implemented and revised their goal setting until we identified the characteristics of effective goals, summarized in the acronym PEERS: Powerful, Emotionally Compelling, Easy, Reachable, and Student-Focused. Powerful goals are goals that make a

socially significant difference in students' lives. Emotionally compelling goals are goals that teachers really want to hit. According to Heath and Heath [33], the best goal is not "just big and compelling; it should hit you in the gut" (p. 76). Easy goals are not watered-down, weak goals, but powerful goals described in a simple manner. Any powerful goal in a classroom involves significant challenges—what might be called "productive struggle." Coaches just need to help teachers identify the most efficient and effective way to hit the goals.

Reachable goals foster hope. Hope has been identified as consisting of three elements: (a) a preferred future or goal; (b) pathways to the goal; and (c) agency, a belief that the goal can be hit [34]. The reachable goals in the instructional coaching model address all three of those elements. First, they are clearly stated and measurable. Second, they include clearly stated pathways to the preferred future by identifying strategies teachers can use to hit the goal. Finally, by stating the goal and identifying strategies, they help teachers see that they can hit their goals and therefore they increase teacher agency.

Finally, student-focused goals make a powerful difference in students' lives, addressing both the students' achievement and engagement needs, especially their need for emotional engagement. Not surprisingly, therefore, interviews with coaches have surfaced where they find teacher-focused goals to be less powerful for sustained change than student-focused goals [25].

Over six years, coaches partnered with University of Kansas researchers to refine a list of questions that coaches could use during coaching sessions. The questions are not a recipe, and each coaching conversation is a back-and-forth interaction, a bit like a ping-pong game. That is, the coach cannot pong until the teacher pings, so to speak, so each conversation looks different, and coaches will ultimately develop their own list of questions. Nevertheless, the following questions provide a starting point for coaches partnering with teachers to set and meet goals.

Identify Questions

*Reality*

What's on your mind?
On a scale of 1–10, with one being the worst lesson you've taught and 10 being the best, how would you rank that lesson?
Why did you give it that number?
Why didn't you give it a lower number?
What pleased you?
And what else?

*Change*

What would have to change to move the lesson closer to a 10?
If you woke up tomorrow, and a miracle happened so that your students were doing exactly what you would like them to do, what would be different? What would be the first signs be that the miracle occurred?" (Solution-Focused Coaching)
If this class was your dream class, what would be different?
What would your students be doing differently if your class was a 10?
Tell me more about what that would look like?
How could you measure that change?
Do you want that to be your goal?
If you could hit that goal, would it really matter to you?

**Application to Visible Learning.** Goal-directed coaching flips the traditional understanding of how professional development proceeds. That is, rather than focusing on the teaching practices at the heart of Visible Learning, instructional coaching, as defined by the Kansas Coaching Project, begins with a powerful student-focused goal identified by the teacher in partnership with the coach. Then the teacher and coach consider strategies the teacher may implement to hit the goal. At this point in the

coaching cycle, the coach suggests strategies for the teacher to consider. When teachers start with a student-focused goal, based on a clear picture of reality, they are often more motivated to hit the goal, and the goal provides an objective standard for assessing effective implementation.

There are numerous powerful, student-focused goals. For example, goals might be that (a) students are able to write a well-organized effective paragraph, as measured by a single-point rubric; (b) students master course vocabulary as measured by weekly tests; or (c) students demonstrate a deep knowledge of core concepts, as measured by short answer responses. The most powerful goals extend beyond units to essential knowledge that can be applied in a variety of settings, and once goals are set, teacher and coach can turn to identifying the most powerful strategy or strategies that can be implemented to meet the goal.

**Identify teaching strategies.** Instructional coaching is about improving instruction to improve outcomes related to both student learning and well-being. Therefore, once a goal has been set, an instructional coach partners with teachers to identify a strategy the teacher will implement to hit his or her goal. Sometimes teachers are able to identify the strategy they want to implement without suggestions from the coach. At other times, when a teacher appears to be searching for suggestions, a coach will share options, usually after asking the teacher a question such as, "Would you like me to share some ideas I've got about strategies you might consider?" [35].

**Application to Visible Learning.** Once the teacher has identified a goal, the coach and teacher collaborate to identify what strategy or strategies the teacher will implement to hit the goal. Most frequently, strategies are those identified as powerful through Hattie's synthesis of meta-analyses [36]. Instructional coaches are most helpful to teachers at this stage of the coaching cycle if they can explain to teachers the effect size of strategies in the playbook, and perhaps even explain effect size itself. For example, if a teacher sets a goal for students' writing, the coach could explain that teacher clarity has an effect size of $d = 0.75$, formative evaluation has an effect size of $d = 0.90$, that feedback has an effect size of $d = 0.75$, and that in combination clarity, formative evaluation, and feedback represent a powerful set of strategies that may be used to hit an achievement goal.

To guide teachers to identify strategies they can use to help students meet goals, coaches can use the following questions, which are largely influenced by the work of Campbell and Van Nieuwerburgh [37].

*Options*

What teaching strategy could you use to hit that goal?
Would you like some suggestions?
What advice would you give someone else if this was their problem? (John Campbell)
Which option gives you the most energy? (John Campbell)

*Next Steps*

What are your next steps?
What can you accomplish this week to move closer to your goal? And what else can you do?
When will you do that?

*4.2. Learn*

**Learn: Explain teaching strategies.** To make it easier for teachers to implement instructional strategies, coaches can enhance and simplify their explanations through the use of checklists [38]. This is not to say that something as complex and artful as teaching can be reduced to a simple checklist. However, when coaches explain a learning structure, such as jigsaw ($d = 1.09$), they can often make it easier for teachers to understand by using a checklist. When they describe strategies using checklists, coaches must be precise, but also provisional, encouraging teachers to make changes or modifications as they see fit, such as the one included in Table 1.

**Table 1.** Checklist: Jigsaw. [26].

| Students Know . . . | √ |
|---|---|
| What group they will be in for the first activity (perhaps by writing down the number for their group). | |
| What group they will be in for the second activity (again, perhaps by writing down the number for their group). | |
| How they are to work together to learn and summarize what they are learning. | |
| The product they need to create to share with the second group. | |
| Before moving to the second group, that what they have created has got their teacher's stamp of approval. | |
| How they should communicate with each other in both groups (in particular, how they should listen and talk). | |
| How they will record (usually take notes or fill out a learning sheet) what they learn from their fellow students in their second group. | |

When coaches share checklists, we have found that they are most effective if they take the partnership approach—that is, their explanations are precise but provisional. If teachers are to implement a practice effectively, they need a clear and precise explanation. However, no one strategy works the same in every unique classroom with every unique teacher and student. Therefore, when coaches describe strategies, we suggest that they ask teachers if they would like to modify the strategy to better meet their students' needs or to better fit their approach to instruction.

In the event that teachers suggest changes that coaches consider ill-advised; coaches can share their concerns in a way that still positions the teacher as the decision maker. For example, a coach who is concerned that students might not have the social skills necessary for jigsaw might say, "One thing I'm wondering is whether or not you think we need to teach the students some interaction strategies before we set up jigsaw? You know your students best, though. I'm curious to know if you think they can succeed at the strategy without learning those strategies?"

When teachers are free to modify the strategies they learn from coaches, there is always the risk that they will make "lethal mutations" [39] changes to a strategy that decreases its impact so significantly that it loses its effectiveness. This is one major reason why setting goals is important. If a teacher does not implement a strategy effectively, the students likely will not meet the identified goal, so the teacher will need to revisit the way the strategy is taught (see the description of the Improve stage below). In short, the goal provides focus for teaching and learning, increases the teacher's and students' motivation, and at the same time, functions as an objective standard of excellence. In most situations, only strategies that are implemented effectively will achieve the desired results.

**Application to Visible Learning.** If coaches are to share Visible Learning, they need to know what Visible Learning is; indeed, they will need to have a deep understanding of it. This is consistent with Hattie's finding that teacher clarity ($d = 0.75$) is an important variable in increasing achievement. One way to deepen, reinforce, and support coaches' knowledge of Visible Learning is for coaches and other educational leaders to create an instructional playbook [40]. A playbook for Visible Learning would be guided by Hattie's synthesis of meta-analyses. An instructional playbook includes: (a) a list of the teaching strategies instructional coaches share with teachers to enable them to hit PEERS goals; (b) one-page summaries of the most important information related to each strategy; and (c) all the checklists coaches would need to describe each strategy.

When creating a playbook for Visible Learning, it is important to keep in mind that Visible Learning is not just a collection of teaching strategies. In fact, Hattie and Zierer [4] make it clear that we misunderstand Visible Learning if we only use the influences at the top of the list while "ignoring those at the bottom of the rankings" (p. xvii) or ignoring that it is "the overlapping of the many factors" (p. xvii) that will have an impact on achievement.

**Learn: Modeling strategies.** In most cases, we learn how to do something better if we see someone else do it before we have to do it ourselves. For this reason, modeling is another important way that coaches help teachers get ready to implement strategies. Interviews with coaches have led to the identification of several ways in which strategies may be modeled [25]. For example, a coach may go into a teacher's classroom and model the strategy in front of the students. In such situations, teachers have reported that they prefer that coaches just model the strategy and not teach the entire lesson; teachers will likely learn more if they observe the model with the checklist in hand. Coaches could also co-teach, with the teacher explaining the content and the coach setting up the activity. For some strategies, coaches can demonstrate the strategy without any students in class. Additionally, the teacher could go to another teacher's classroom to see how to implement jigsaw by watching another teacher implement the strategy, or the teacher could watch video of a teacher using the strategy. Sometimes more than one of these versions of modeling is used.

**Application to Visible Learning.** As is the case with explaining strategies, coaches need to have a deep understanding of Visible Learning in order to model strategies. Teachers who see a teaching strategy taught a certain way will naturally attempt to teach it that way. Our research suggests that the use of video is an extremely powerful tool for coaches to assess how effectively they are implementing practices, and that is consistent with Hattie's finding that micro-teaching with video ($d = 0.88$) is a high-impact strategy.

Coaches also need to ensure that the alternatives to in-class modeling (such as visiting a teacher implementing a strategy or watching a video of a teacher implementing a strategy) provide excellent demonstrations of Visible Learning. Coaches can foster and support high-quality implementation of Visible Learning by identifying teachers who are successfully implementing this construct or by building a library of videos of teachers effectively demonstrating Visible Learning practices.

*4.3. Improve*

**Improve: Making adaptations to meet goals.** The first time a teacher implements a strategy, it usually does not lead to the students meeting the goal. For that reason alone, coaches play an essential role in supporting implementation because they can encourage and support the teacher when he or she might otherwise be about to give up on a given strategy. During each conversation in the Improve stage, coaches should begin by confirming that they and the teacher are concerned with the same issues. Coaches might ask questions such as "What's on your mind?" [41] and "Given the time we have today, what's the most important thing for us to talk about?" [42].

Following this, the coach and teacher should discuss what progress is being made toward the goal. The coach or teacher might assess student writing with the help of a rubric to see how close the students are to the goal; teacher and coach might also discuss how effectively various teaching strategies, such as worked to ensure students are learning the content they need to write effective paragraphs. To monitor progress, coaches could ask questions such as the following.

*Review Progress*

What has gone well?
What are you seeing that shows this strategy is successful?
What progress has been made toward the goal?
What did you learn?
What surprised you? (Steve Barkley, personal conversation, 2009)
What roadblocks are you running into?

It is rare for a goal to be met without some adaptations. In most cases, the coach and teacher will need to do some creative problem solving to meet the goal. For example, the teacher might explain that he or she needs some additional strategies to increase student knowledge either for variety or because jigsaw is not empowering students to meet the goal. The teacher or coach might then come up with other strategies such as concept diagrams or peer coaching to support student learning.

In some cases, teachers might determine that they have the correct strategy but recognize that the way the strategy is being used needs to be modified to be effective. This might happen, for example, when a teacher has dramatically modified how a strategy is being implemented and then realizes that the way he or she teaches it needs to be closer to how it was described on the original checklist.

In addition, teacher and coach may determine that the goal needs to be changed. After looking at students' writing, the teacher might determine, for example, that 80% of the students getting five out five correct was too conservative a goal and decide instead that the goal should be 90% of the students get five out five on the rubric three times in a row. Additionally, the teacher, in conversation with the coach, might decide that the rubric needs to be modified to better assess student performance.

One final option is to determine that no adaptations are needed. Even though the goal has not been hit, the coach and teacher may determine that if they continue to use the strategy, it will eventually lead to the goal without modification. Sometimes no change is the best way forward.

The last part of the Improve stage is for the coach and teacher to plan next steps. In general, it is better to over- than to under-plan. Teaching is complex work, and plans can easily get lost in the midst of everything else that teachers need to accomplish. Effective planning makes it much more likely that strategies will be implemented. Some questions coaches can ask to promote careful planning are listed below.

*Plan Next Actions*

When should we meet again?
What tasks have to be done before we meet?
When will those tasks be done?
What will you do next in pursuit of your goal?
On a scale of 1–5, how committed are you to this goal now?

**Application to Visible Learning.** During the Improve stage, coaches are more effective if they can suggest additional elements of Visible Learning to support teachers in helping their students meet goals. For example, a teacher who implements jigsaw to help students master core concepts in a mathematics class may find that he or she needs to implement classroom management strategies for jigsaw to work effectively [43]. Coaches need to ensure that the suggestions they make during the Improve stage are appropriate for each teacher and his or her students. For example, the effect size for time on task is not as robust as other practices, but it could be very important in a classroom where time on task is less that 60%.

Finally, the Improve stage is an opportunity for coach and teacher to explore how the various components of Visible Learning may be combined to increase impact. Although teachers cannot learn all elements of Visible Learning at one time, the more they implement more of the factors identified in the 10 mindframes summarized above, the greater impact they will have on student learning.

Once a cycle has been completed, coach and teacher can begin a new cycle if time permits. Over time, the teacher will learn more about Visible Learning by implementing it; that is, by using the approach to help their students meet powerful goals. Ultimately, the goal is for the teacher to be able to move through the cycle—analyzing reality, setting goals, identifying and learning strategies, and making modifications—with minimal help from the coach. When teachers learn to draw on all of the factors embodied in the 10 mindframes identified by Hattie and Zierer [4], Visible Learning will stop being merely an idea and truly become a reality.

## 5. Conclusions

Instructional coaching is not a strategy-specific approach to professional learning. Any teaching strategy or instructional program may be implemented through instructional coaching [44]. What matters is that teachers learn new strategies in the context of their day-to-day work, helping students attain higher achievement and greater well-being. When teachers work on powerful goals that matter to them, and they partner with coaches who have a deep understanding of the strategies being learned

and who provide adaptive support for professional learning when needed, research moves from being a "nice" idea to being a central part of what happens in schools. And when powerful research like Visible Learning is implemented widely and successfully, students are much more likely to succeed.

This paper is intended to open the door to further study of instructional coaching and Visible Learning. It is an exploration of how instructional coaching could be employed to support implementation of Visible Learning, but further studies are needed on the particular aspects of instructional coaching and Visible Learning. Specifically, further study needs to be conducted on the impact instructional coaching can have on implementation of Visible Learning and the impact that instructional coaching and Visible Learning, in tandem, can have on student achievement.

More focused studies addressing micro-elements of coaching are also recommended. For example, studies that address differences in implementation between goal-directed coaching and coaching focused on implementation support or studies that explore the impact of instructional playbooks on teacher implementation would be very helpful.

A major concern that needs serious consideration is a challenge at the heart of any form of professional development for Visible Learning. That is, the construct of Visible Learning is an integrated approach to instruction, captured in the 10 mindframes, rather than a list of things to try one-by-one. However, the research conducted utilizing lean-design research suggests that teachers are more likely to adopt and continue to implement strategies when they want to meet a student-focused goal. Therefore, thought and study needs to be given to how to share all of Visible Learning in a way that is manageable and a part of goal-directed learning.

The research conducted at the Kansas Coaching Project has made one thing clear: Supporting teachers in their implementation of evidence-based practices is much more complex than simply holding a workshop and expecting teachers to implement certain practices. Professional development that fosters genuine professional learning and leads to real improvements in the classroom has to position teachers as partners, and be job-embedded, explicit, and adaptive.

**Funding:** This research was funded by GEAR UP, 84.334A; 84.334S, a grant program designed to increase the number of low-income students who are prepared to enter and succeed in postsecondary education. It was also funded by the Institute of Education Science (IES), whose mission is to provide scientific evidence on which to ground education practice and policy.

**Conflicts of Interest:** The author declares no conflict of interest.

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
