# Peer review of "Instructional Coaching for Implementing Visible Learning: A Model for Translating Research into Practice"

_education, doi:10.3390/educsci9020101_

Round 1

Reviewer 1 Report

This is a well crafted piece.  there is just one proofreading comments I would make - see 248 and 251 for repetition of 'finally'.

My concern sits in the purpose of the paper.  The paper advocates for instructional coaching to be used in the implementation of visible learning.  Yes, instructional coaching may well be ideally placed to support teacher's implementation of VL, though a stronger position would be to show that through evidence of that practice.

It also reads as a formulaic 'how to' for educational consultants to act as fly in coaches.  I would prefer to read an exploration for how teachers might engage in practitioner inquiry for 'intelligent problem solving' (Dewey) as Hattie suggested as the purpose in his initial rendering of VL rather than formulaic adoption of high impact activities.  the latter having been a significant detriment to the real potential of Vl that has largely been lost on teachers and schools.

While attention is brought to issues in transposing and the model presented gives scope to reach intelligent problem solving through teachers engaged as participant researchers in their own practice, there is also scope for Instructional coaching as presented to be formulaic.  This is particularly the case when framed in terms of success measured be "more likely to implement", suggesting routinised take up rather than intelligent problem solving.  this is exactly what has been so detrimental in the response to Hattie's work and the opposite of what he initially aimed for.

Author Response

I’m grateful for the reviewer’s comments, and where possible, I’ve done my best to respond with changes.

Item one: The repetition of finally has been revised. Thank you for finding that.

Item two: “The paper advocates for instructional coaching to be used in the implementation of visible learning.”  I intended a slightly different purpose for the paper, which was to explain how instructional coaching can be used to support implementation of effective instructional models or constructs, using visible learning as an example. As such, I wrote the paper to summarize our research on instructional coaching, and then explain how our findings could be applied to visible learning.  I have tried to clarify this in the revised paper. And I certainly agree that studies specifically focussed on instructional coaching and visible learning are needed. My understanding though was that I was invited to write the paper with the purpose I submitted.  

Item three: “It also reads as a formulaic ‘how to’ for educational consultants.” I’m grateful for this feedback because that is the opposite of what we have found about instructional coaching, but I can see how the paper might give that impression. I’ve tried to address this by revising the paper to include a clarification in the body of the text, that coaching is what I have previously referred to as an informed and adaptive response. I’ve also included the passage below.  I’d be grateful feedback on whether this is sufficient to position the reading of the rest of the paper so readers see that instructional coaching as it is described here is not a formulaic model, but in fact more like the intelligent problem solving mentioned by the reviewer.

Instructional coaching, although laid out sequentially in this paper, is not a simplistic one-size fits all formula for improvement. Instructional coaches respond to the context in which coaching occurs, shaping what they do based on students’ needs, teachers’ insights, and other important factors.  This approach has been described as an informed-adaptive approach (XXXX, 2011). 

Informed coaches know a lot about the situation where coaching occurs.  They have a deep understanding of effective instruction, individual teachers’ strengths and concerns, and students’ unique characteristics. They are emotionally intelligent which means they are skilled at fostering trust and building relationships that are more likely to lead to learning. Informed coaches have a deep knowledge of instructional practices which enables them to offer more options to teachers who partner with them to meet students’ needs.  

Adaptive coaches respond to the unique contexts in which coaching occurs.  Although instructional coaching, as it is described here, occurs within a framework, that framework is not a formula to be followed by rather a container for coaching conversations. Thus, each coaching conversation is individualized to each context.  The questions coaches ask, the goals that are set, the teaching practices chosen by teachers, the way teachers learn and practice new strategies, and the modifications that are made are all unique to the partnership between teacher and coach. Instructional coaching involves a structure, but in action it is individualized process, uniquely co-constructed by each coach and teacher. 

Throughout the revised paper I have tried to emphasize the collaborative nature of the coach and teacher’s partnership, which I believe is similar to the intelligent problem solving you propose. I hope these revisions make it clearer that this is not a formulaic approach to coaching.

Item four.  “I would prefer to read an exploration for how teachers might engage in practitioner inquiry for ‘intelligent problem solving’ (Dewey).  Whether or not intelligent problem solving would be superior to instructional coaching for improving student achievement is an interesting and important question that could be addressed empirically.  Unfortuately, I can only report on the research we have conducted, which shows great promise for supporting implementation of Visible Learning. As noted above, I also feel that instructional coaching is closer to “intelligent problem solving” than maybe was apparent in the earlier version of this paper, and I have attempted to address this issue through out the paper.

I have attached a revised version of the paper.

Thank you

Reviewer 2 Report

The paper does not have scientific format and does not provide any novelty for the ‘Visible Learning’ field. Therefore, I recommend rejecting it for publication in “Education Sciences Journal”.

For the text review, I suggest to include the conclusions of the study in the abstract. Apart from that, what is the title of the first section? In this item authors should to establish the context of the work being reported. It is important defines the presents the background that supports the study and also details the usefulness of the work.

Illustrations are the most efficient way to present your results. All the figures are too big in this paper. It is recommended to think about adjust them, to include the graph axes and to put information about sources.

Write a clear conclusion. This section must to shows how the work advances the field from the present state of knowledge.

Author Response

I’m grateful for the reviewer’s comments, and I will do my best to reply.

Item One: “The paper does not have a scientific format.”  This is correct, but when I was asked to write this paper I assumed that my task was to discuss how the twenty years of research on instructional coaching could be applied to Visible Learning, not that I was to describe a single research study, such as the many referenced in the research section.  I’m not quite sure how to resolve this issue, but it is clear that my goal for the paper is different that the reviewers’ expectations. Unfortunately, this different understanding of the purpose of the paper will make it difficult for me to comply with all suggestions.  The scientific paper format suggested here is not appropriate for this paper because this paper is not a report on a scientific study, but rather an exploration on how many research studies could inform how Visible Learning is implemented in schools.

Item Two: “The paper does not provide any novelty for Visible Learning field.”  Instructional coaching is one of the most important approaches to professional development in North American and increasingly around the world.  To date, there has never been a paper that discusses research on instructional coaching and Visible Learning or indeed any paper writing about how instructional coaching can be used as a methodology for sharing Visible Learning.  I feel this paper is a very important contribution to the study of Visible Learning because it is the first paper to ever explore how coaches could support implementation of Visible Learning or indeed any instructional construct. For Visible Learning to be implemented widely, a model for implementation similar to instructional coaching is needed.

Item Three: “include the conclusions of the study in the abstract.” I have added the following sentence: Instructional coaching shows great promise as an approach for supporting implementation of Visible Learning, but more study is needed.” 

Item Four: “What is the title of the first section?”  I have labeled this Introduction, and revised the introduction to clarify the importance of this topic.

Item Five: “All the figures are too big in this paper… adjust them to include the graph axes and to put information about sources.” All of the figures were reduced in size, the graph axes have been labeled, and all sources for figures are included.

Item Six: “Write a clear conclusion [showing] how the work advances the field from the present state of knowledge.”  The conclusion has been expanded, but again, since this paper is not a scientific paper per se, but a summary of scientific papers, and an exploration of how they apply to Visible Learning, the conclusion is not exactly the format of a scientific paper.

I have attached a revised version of the paper. There were so many changes that it was impossible to read the paper clearly when it was in track changes format, so I'm sending a clean version. If you would like to see the paper with track changes, I can send that upon request.

Reviewer 3 Report

This is a very thoughtful review and I very much liked the suggested formats.

This manuscript contains new information that I believe will be useful as people attempt to integrate VL into their schools. 

The author makes a compelling case about the role of coaching and provides a strong model to do so.  

Author Response

Thank you for your comments.

Round 2

Reviewer 2 Report

Ok, accept in present form.
